# Effect of the Administration of a Lyophilised Faecal Capsules on the Intestinal Microbiome of Dogs: A Pilot Study

**DOI:** 10.3390/genes14091676

**Published:** 2023-08-25

**Authors:** Sandra Carapeto, Eva Cunha, Isa Serrano, Pedro Pascoal, Marcelo Pereira, Raquel Abreu, Sara Neto, Beatriz Antunes, Ricardo Dias, Luís Tavares, Manuela Oliveira

**Affiliations:** 1CIISA—Centro de Investigação Interdisciplinar em Sanidade Animal, Faculdade de Medicina Veterinária, Universidade de Lisboa, Av. da Universidade Técnica, 1300-477 Lisboa, Portugalmoliveira@fmv.ulisboa.pt (M.O.); 2Associate Laboratory for Animal and Veterinary Science (Al4AnimalS), Av. da Universidade Técnica, 1300-477 Lisboa, Portugal; 3BioISI—Biosystems & Integrative Sciences Institute, Faculdade de Ciências, Universidade de Lisboa, Campo Grande, 1749-016 Lisboa, Portugalrpdias@ciencias.ulisboa.pt (R.D.); 4Canil Municipal de Peniche, Câmara Municipal de Peniche, Rua Doutor Ernesto Moreira G, 2520-428 Peniche, Portugal; 5cE3c—Centre for Ecology, Evolution and Environmental Changes & CHANGE—Global Change and Sustainability Institute, Faculdade de Ciências, Universidade de Lisboa, Campo Grande, 1749-016 Lisboa, Portugal

**Keywords:** faecal microbiota transplantation, intestinal microbiome, dogs, diarrhoea, faecal capsules

## Abstract

Faecal Microbiota Transplantation (FMT) is a promising strategy for modulating the gut microbiome. We aimed to assess the effect of the oral administration of capsules containing lyophilised faeces on dogs with diarrhoea for 2 months as well as evaluate their long-term influence on animals’ faecal consistency and intestinal microbiome. This pilot study included five dogs: two used as controls and three with diarrhoea. Animals were evaluated for four months by performing a monthly faecal samples collection and physical examination, which included faecal consistency determination using the Bristol scale. The total number of viable bacteria present in the capsules was quantified and their bacterial composition was determined by 16S rRNA gene sequencing, which was also applied to the faecal samples. During the assay, no side effects were reported. Animals’ faecal consistency improved and, after ending capsules administration, Bristol scale values remained stable in two of the three animals. The animals’ microbiome gradually changed toward a composition associated with a balanced microbiota. After FMT, a slight shift was observed in its composition, but the capsules’ influence remained evident during the 4-month period. Capsules administration seems to have a positive effect on the microbiota modulation; however, studies with more animals should be performed to confirm our observations.

## 1. Introduction

The microbiota of the gastrointestinal tract comprises a vast number of microorganisms [1]. The constitution of its bacterial population is crucial, since its stability and proper composition are beneficial for the maintenance of the host’s health. These microorganisms participate in several metabolic pathways, such as the production of short-chain fatty acids and secondary bile acids, which are important for the immune system, maintenance of the intestinal barrier, and resistance to colonisation by pathogenic bacteria [2]. Age, diet, genetics, and many other environmental factors can play a significant role in maintaining a healthy microbiome; however, the changes that these factors may cause are low when compared to those seen in diseased animals. Acute or chronic gastrointestinal inflammation often leads to dysbiosis, which is characterised by significant reductions in microbial diversity compared to healthy animals [3]. Dysbiosis is marked by broad changes in the composition of microbial communities, decreased species diversity, alterations in the relative abundance of specific organisms, and consequent shifts in the production of metabolites. However, the possibility that a healthy animal may present an unstable microbiota should not be overlooked [3,4].

If dysbiosis occurs, it is necessary to control the instability of the intestinal microbiota. Diet, prebiotics, probiotics, symbiotics, and antimicrobials are frequently used with this purpose, but in some cases, these treatments may not be effective, or, in the case of antimicrobials, their administration may be associated with increased antimicrobial resistance or increased dysbiosis [5]. Faecal Microbiota Transplantation (FMT) aims to restore intestinal stability by introducing a healthy “microbial ecosystem” in the host [1]. FMT consists of transferring faecal material from a healthy donor into the gut of a non-healthy recipient to restore its gut microbiota. The beneficial effects of this alternative method have not yet been clarified. However, it is known to contribute to the enrichment of the microbiome and to the alteration of microbial profiles [2].

In veterinary medicine, FMT has been considered a possible treatment in cases of giardiasis refractory to treatment, chronic enteropathy, parvovirus infection, and other types of acute diarrhoea, such as haemorrhagic gastroenteritis [5,6,7,8,9]. To date, no serious adverse effects associated with FMT have been reported, which is probably due to limited data availability [10]. It should be noted that currently, both in human and veterinary medicine, there is still no consensus on the methods of faecal material preparation and storage, donor selection criteria, doses to be administered, time interval between administrations, and the frequency and route of administration [2].

The goal of this pilot study was to evaluate the influence of the long-term administration of freeze-dried faecal capsules via oral route on the composition of the intestinal microbiota of dogs and their effectiveness in correcting animals’ faecal consistency.

## 2. Materials and Methods

### 2.1. Animals in the Study

For this study, five animals were selected from an official rescue institution. All animals were cared for according to the rules given by the current EU (Directive 2010/63/EC) and national (DL 276/2001 and DL 113/2013) legislation and by the competent authority (Direção Geral de Alimentação e Veterinária, DGAV) in Portugal. The ethical guidelines of the official rescue centre were followed in this study, and a written protocol was established to perform the study. All procedures were performed by trained veterinarians, and only non-invasive procedures were used.

From the five study dogs, two presented a normal faecal consistency and were selected as the positive (PC—subjected to capsules administration) and negative (NC—to which no capsules were administered) controls; the other three animals presented chronic diarrhoea (over or equal to 3 weeks [11]) and constituted the target animals of the study (treatment group).

The selected dogs were housed in a kennel in separate outdoor covered boxes (except for Animal 1 and the NC, which were together in the same box), shared a common environment (including water and air quality and cleaning procedures), and were fed with the same diet. These dogs exhibited the characteristics outlined in Table 1, which includes the classification of their initial faecal consistency according to the Bristol Stool Scale [12]. The use of this scale was due to its simplicity, not being exclusive to human medicine [13].

The animals were of indeterminate breed, aged 2 years or older and with a body weight of more than 9 kg and less than or equal to 27 kg. The selected dogs did not receive any type of medication in the six months before the study, excluding internal and external dewormers. Animals’ vaccination status was regularised, except for Animal 2, which had only recently entered the kennel. Animals presented no gross signs of disease apart from the alteration in faecal consistency in Animal 1, Animal 2, and Animal 3.

### 2.2. Faecal Microbiota Transplantation

Oral FMT was performed through the administration of oral capsules commercialised since 2016 by *AnimalBiome*^®^ (Oakland, CA, USA). According to the information available at the *AnimalBiome*^®^ website, capsules are produced through the lyophilisation of faecal material from a rigorously selected canine donor, aiming to exclude the presence of pathogens or multidrug-resistant bacteria within the transplanted faecal material. Donor animals are preferably aged between 1 and 10 years, with optimal body condition and normal faecal consistency, and sound health upon physical examination. Moreover, with the absence of antimicrobial usage within the last 3, 6, or 12 months, up-to-date vaccinations and empirical deworming with broad-spectrum drugs are further prerequisites. Laboratory screenings demand normal haematology and biochemistry, negative faecal flotation results, and negative results for pathogens such as Giardia and diseases like parvovirus and distemper [2,14,15]. In this case, considering that it involved a commercial product, no information was obtained regarding the faecal donor of the capsules utilised during the study.

Regarding the composition of the capsules, in addition to the faecal material, these also contain inactive ingredients such as glycerol and the enclosing capsule, which confers resistance to enzymatic action within the gastric and intestinal compartments.

### 2.3. Experimental Design

This pilot clinical trial was a controlled study that included one dog as a negative control (with normal faecal consistency and no treatment) and two test groups: one with a single animal showing normal faecal consistency (positive control) and another group with three animals demonstrating altered faecal consistency (treatment group). At the beginning of the trial, and before any capsule’s administration, samples were collected from each animal that participated in the study, which were used for comparison along the clinical trial. Considering that, these samples were used as controls of the animals from which they were originated.

The animal of the positive control and the treatment group were submitted to the oral administration of one capsule of *AnimalBiome*^®^ per day for two months (60 days). The negative control animal was not submitted to any treatment or placebo. The study period comprised a total of four months and included the monthly collection of a faecal sample per animal (Figure 1). The first sampling was carried out at the beginning of the study (T0) without any capsule being administered to the animals, as previously mentioned. The second faecal sampling was performed 30 days after the first administration of the freeze-dried capsules (T1). The third sampling was performed 30 days after T1, at the end of the FMT administration (T2). The last sampling (T3) was performed one month after T2, aiming to assess the long-term efficacy of the faecal capsule’s administration.

At each sampling moment, a physical examination of each one of the animals was performed, including the general observation of the animal, faecal consistency evaluation, measurement of rectal temperature, respiratory and heart rate, cardiac and respiratory auscultation, evaluation of the eyes, ears, nose, mouth, skin for any abnormalities, observation and palpation of lymph nodes, abdominal palpation and testicles, and observation of the mucous membranes. In addition, two months after the end of the capsule administration, a new physical evaluation was performed (T4).

In addition to the monthly visits, weekly telephone calls were made to ascertain the state of health of the animals and the presence or absence of side effects.

### 2.4. Collection, Packaging and Storage of Faecal Samples

Faecal samples were collected using sterile plastic cups after defecation either during the animal’s daily walk or after their feeding period. The samples were temporarily stored in a Styrofoam box with freeze plates and later moved to a −20 °C freezer for long-term storage within 24 h.

### 2.5. Quantification of the Bacterial Population Present in the Capsules

First, 1 mg of the faecal material present inside a capsule was diluted in 1 mL of saline solution. Afterwards, the quantification of the total number of viable bacteria present in the lyophilised capsules was performed by inoculating 100 µL of serial 10-fold dilutions up to the 10^−10^ dilution onto the surface of a Trypticase Soy Agar (TSA) plate, in duplicate. All plates were incubated aerobically at 37 °C for 24 h, after which the number of colonies present on each plate was quantified.

### 2.6. Characterisation of Capsules and Faecal Samples Microbiome

To determine the composition of the microbiome present in the capsules and faecal samples, DNA extraction was performed using the QIAamp^®^ PowerFecal^®^ Pro DNA kit [16]. The DNA was then amplified by PCR using a Biometra T1 Thermocycler T-1 Thermoblock. The composition of the master mix used was the following: 60 mM Tris-SO_4_, 20 mM (NH_4_) 2SO_4_, 2 mM MgSO_4_, 0.3 mM dNTPs, 3% Glycerol, 0.06% IGEPAL^®^ CA-630, 0.05% Tween20, 125 units/mL LongAmp Taq DNA Polymerase; pH 9.1 at 25 °C and 1.5 µL of each primer, 27f and 1492r, targeting the 16S rRNA. The amplification protocol included 35 cycles as follows: denaturation (95 °C, 10–30 s), annealing (55 °C, 15–60 s) and replication (65 °C, 50 s). Subsequently, amplification products were visualised through gel electrophoresis and purified using the Solid-Phase Reversible Immobilisation (SPRI) technique with magnetic beads [17,18]. After, a DNA library was built. The samples were then quantified using the Qubit^®^ fluorometer, and the adapters were fixed for subsequent genomic sequencing, which was performed using GridION X5, commercialised by Oxford Nanopore Technologies^®^ [19].

### 2.7. Statistical Analysis and Data Handling

Samples were analysed using a custom analytical pipeline developed by BioISIGenomics^®^ to obtain a highly accurate taxonomic classification. Initially, for each sample, operational taxonomic units (OTUs) were identified and, subsequently, a heatmap was established to compare the bacterial composition of the samples. In addition, each sample was analysed for its α diversity based on Shannon’s diversity index and between samples for their β diversity based on the Bray–Curtis dissimilarity index. The β diversity was expressed using Principal Coordinates Analysis (PCoA). The Kruskal–Wallis non-parametric statistical test was applied in multiple paired comparisons to compare the α diversity of each sample (*p*-value = 0.05).

## 3. Results

### 3.1. Animals’ Physical Examination

During the study period, the physical examination of the animals revealed no mild or gross signs of disease, and no side effects associated with the administration of the faecal capsules were reported.

Concerning faecal consistency, after 30 days of capsule administration, there was an improvement in this parameter (Table 2). In fact, according to the weekly check-up, diarrhoea resolution occurred in the third week after the beginning of FMT. However, Animal 2 did not reach level 4 of faecal consistency as established by the Bristol Scale and presented a level 5 faecal consistency at T1. At T2, all the animals under study maintained an improved faecal consistency. In the months following the end of capsule administration (T3 and T4), only Animal 2 had a regression in the faecal consistency level, although not reaching the level observed in the beginning of the study. Throughout the study, the dogs selected as controls continuously exhibited faeces with normal faecal consistency.

### 3.2. Microbiome Analysis

To analyse the results obtained by genomic sequencing, the composition of the capsules and the healthy intestinal microbiota were determined first. Subsequently, the analysis of the remaining faecal samples was performed to evaluate the influence of capsule administration on the concentration of each one of the bacterial groups present in the faecal microbiota of the animals under study.

#### 3.2.1. Capsules

The capsules presented a total number of viable bacteria of 10^8^ CFU/mL. Through genomic sequencing, it was possible to observe that most of the bacteria present in these capsules belonged to the phylum Firmicutes (96.56%), which was followed by the phylum Actinobacteria (2.27%) and then the phylum Bacteroidetes (1.13%). At the class level, Clostridia was the most predominant (86.49%), which was followed by Bacilli (9.04%) and by Bacteroidia, Coriobacteriia and Erysipelotrichi classes present in a lower relative concentration (1.13%, 2.27% and 1.06%, respectively). The most frequent bacterial families were *Lachnospiraceae*, *Clostridiaceae* and *Peptostreptococcaceae* (41.13%, 23.13% and 20.38%, respectively). At the species level, a higher concentration of *Clostridium hiranonis* (38.28%), *Blautia* spp. (29.24%), *Ruminococcus gnavus* (6.76%) and *Streptococcus luteciae* (8.09%) was observed. The presence of *Clostridium perfringens* (4.80%) at a higher concentration than *Faecalibacterium prausnitzii* (1.04%) should be highlighted (Figure 2).

#### 3.2.2. Controls

In this study, the initial faecal samples (T0) of the dogs selected as controls (positive and negative control) were evaluated to determine the composition of the gut microbiome in an equilibrium state, in the absence of diarrhoea, and without the administration and possible effect of the faecal capsules. In the case of the negative control (NC), as this animal was not subjected to the administration of FMT capsules, the data obtained from the samples collected in the following months were also considered to act as negative control.

In these animals, Firmicutes (93.68%) predominated in all faecal samples, which was followed by Bacteroidetes (5.65%), while Fusobacteria, Proteobacteria and Actinobacteria were less prevalent (0.09%, 0.09% and 0.48%, respectively). Regarding bacterial classes, there was a higher concentration of Clostridia, Bacilli and Bacteroidia (20.82%, 72.86% and 5.66%, respectively) in contrast to the low concentration of Gammaproteobacteria and Actinobacteria (0.02% and 0.003%, respectively). At the family level, a high number of bacteria from the *Lactobacillaceae* were observed, while the *Enterobacteriaceae* was one of the least represented. The highest concentrations of *Clostridiaceae*, *Peptostreptococcaceae* and *Lachnospiraceae* were observed in sample T1 from the NC animal (19.97%, 26.38% and 23.16%, respectively). More specifically, the predominant species in most faecal samples was *Lactobacillus reuteri* (24.50%), which was followed by other species belonging to the genus *Lactobacillus*. Most of the samples presented considerable amounts of *Clostridium hiranonis* and *Blautia* spp., while *Streptococcus* spp. and *Faecalibacterium prausnitzii* were found in lower concentrations. It was also possible to observe the commensal presence of *Clostridium perfringens* (0.48%).

Regarding the NC animal, it is important to mention an evident change in composition in the T1 sample, which was characterised by a significant increase in the Clostridia class, followed by a subsequent decrease in the Bacilli class, and an increase in the diversity of bacterial families.

#### 3.2.3. Microbiome of Animals Subject to FMT

##### 
Phylum


Initially, all dogs showed a higher concentration of Firmicutes and Proteobacteria and a lower concentration of Bacteroidetes and Actinobacteria compared to those present in the freeze-dried capsules (Figure 3).

During FMT, a decrease in the concentration of Firmicutes was observed in all faecal samples, together with an increase in Bacteroidetes, except in the T2 sample from the positive control (PC) in which the opposite was observed. The concentration of Actinobacteria throughout FMT administration increased, except in Animals NC and 2. Finally, slight changes in Proteobacteria prevalence were observed at T1. However, at T2, a higher concentration of this phylum was evident in the samples from Animals 2 and 3.

After FMT, Firmicutes and Bacteroidetes variation was opposite to the one observed during transplantation, excluding Animal 2. Actinobacteria decreased markedly in Animal 3, while Proteobacteria increased in Animals 2 and 3. In the faecal samples taken at T3, Firmicutes remained predominant, which was followed by Bacteroidetes. However, when compared to T0, Firmicutes had a lower concentration and Bacteroidetes had a higher one. At T3, the percentage of Actinobacteria was not quite different from that at T0. Finally, when compared to the first faecal samples analysed, Proteobacteria had a clear increase in Animals 2 and 3.

##### 
Class


At T0, Bacilli predominated in all faecal samples, showing higher values than those present in the FMT capsule, while Clostridia, Bacteroidia, Coriobacteriia and Erysipelotrichi showed lower values (Figure 4).

During transplantation, Bacilli concentration was lower than at T0 in all samples except for Animal 2. The opposite was observed for Clostridia, Erysipelotrichi and Coriobacteriia, except for Animals 2 and NC. On the other hand, Bacteroidia increased in all animals during FMT administration.

The samples collected at the end of the study showed a marked increase in Bacilli except for those from Animals 2 and PC, although the decrease present in the animal selected as PC was not evident. Additionally, Clostridia and Coriobacteriia showed a decrease in all the animals under study except in samples from the dogs selected as controls, in which an increase was observed. Furthermore, Bacteroidia decreased in all dogs except in Animal 2, while Erysipelotrichi decreased in Animal 2. Compared to T0, Bacilli and Clostridia concentration decreased and increased in all samples, respectively, excluding for those of canid NC. Bacteroidia presented a higher concentration in all dogs at the end of the study and so did Erysipelotrichi except for Animal 2. Finally, at T3, Coriobacteriia were present at similar levels than those initially obtained.

##### 
Order


In terms of variation at the order level throughout the study, they were like those observed at the class level. *Lactobacillales*, *Clostridiales*, *Bacteroidales*, *Coriobacteriales* and *Erysipelotrichales* showed a similar variation to the classes Bacilli, Clostridia, Bacteroidia, Coriobacteriia and Erysipelotrichi, respectively.

##### 
Family


At T0, *Lactobacillaceae* and *Enterobacteriaceae* were present in higher concentration in all faecal samples when compared to the capsules. *Streptococcaceae* was also in higher concentration in the samples from Animals 3 and PC. In contrast, *Clostridiaceae*, *Lachnospiraceae*, *Peptostreptococcaceae* and *Prevotellaceae* were present at a lower concentration in the faecal samples when compared to the one from the freeze-dried capsules (Figure 5).

During FMT, *Lactobacillaceae* decreased in most faecal samples, apart from T1 samples from Animal 2 and the T2 samples from Animal PC. Most faecal samples showed a higher concentration of *Clostridiaceae* compared to the initial sample, except for those from Animals 2 and NC. When comparing the *Lachnospiraceae* and *Prevotellaceae* composition in the first faecal samples with those obtained during FMT, all dogs samples showed a higher concentration of these bacterial families, although at T2, only Animals 2 and 3 showed an increase in *Lachnospiraceae,* and Animal PC did not present an increase in *Prevotellaceae*. *Peptostreptococcaceae* presented a variation similar to *Lachnospiraceae*; however, compared to T0, different dogs showed divergent results. *Streptococcaceae* exhibited different modifications among the animals’ samples obtained at T1; however, at T2, all dogs subjected to FMT showed a lower concentration of this family when compared to T0. Finally, *Enterobacteriaceae* presented lower concentrations compared to those present in T0 samples from all dogs, except for Animal NC.

After FMT, *Lactobacillaceae* presented divergent variations; however, by the end of FMT, all the samples from the animals under study presented lower values than those from the sample taken at T0. Additionally, *Clostridiaceae* and *Peptostreptococcaceae* presented different variations; however, when compared to T0, *Clostridiaceae* presented a higher concentration in samples collected during the assay except in samples from Animals 2 and NC, while *Peptostreptococcaceae* were only present at a higher concentration in the samples from Animal 3. At T3, *Lachnospiraceae* and *Prevotellaceae* decreased except for Animal 2. However, when comparing to T0, all animals showed higher values of these families, except the NC animal. Furthermore, in samples from T3, *Streptococcaceae* showed lower values than those obtained at T0 in all dogs with the NC animal having the lowest concentration of this bacterial group, as it was the only one that showed a reduction in *Streptococcaceae* concentration at T3. Finally, at T3, different variations in *Enterobacteriaceae* were observed between animals; however, all faecal samples showed lower values compared to T0 samples.

##### 
Genus


Regarding bacterial genera, the differences observed in the analysed faecal samples were like those mentioned above for the family level.

##### 
Species


At the beginning of the study, all dogs showed a reduced concentration of *Blautia* spp., *Clostridium hiranonis* and *Faecalibacterium prausnitzii* compared to the one present in the freeze-dried capsules. In addition, a lower concentration of *Streptococcus* spp. was also observed in the samples from all dogs except for those of Animals 3 and PC (Figure 6).

During administration, *F. prausnitzii* and *Blautia* spp. concentrations were higher than at T0, although the increases in Animals 2 and 3 samples taken at T1 were relatively unremarkable. However, the samples from these two animals were the only ones that showed an increase in these two bacterial species at T2. As far as *C. hiranonis* is concerned, the concentrations obtained at T1 and T2 did not reach the low values present at T0 except for samples obtained at T1 and T2 from Animal 2 and at T2 from Animal NC, with the latter presenting the lowest value of *C. hiranonis* at T2. Finally, after 30 days of FMT, *Streptococcus* spp. values diverged among faecal samples. However, in the second month, there was a lower concentration in comparison to T0, except for the samples from Animal NC.

After the end of the FMT, the variation observed in *F. prausnitzii* diverged among animals, but samples from all dogs showed a higher value than the original one, although the increase was less evident in the NC animal. One month after the end of FMT, *Blautia* spp. decreased in all dog samples; however, samples from all dogs that were submitted to capsule administration showed higher values at T3 than those present at T0. *C. hiranonis* prevalence had divergent variations among animals, but all faecal samples showed an increase in this bacterial group in comparison to the levels present in the beginning of the study, except in those from Animals 2 and NC. Finally, *Streptococcus* spp. increased in all samples except for the one from the NC animal, which presented the lowest value compared to the other samples obtained at T3. However, all samples showed lower values of this bacterial group when compared to T0 samples, being less evident in the NC animal.

#### 3.2.4. Comparation between Animals

To compare the bacterial populations present in each faecal sample, a heatmap was performed to check the similarity level between the samples obtained from different animals and different timepoints. Results were divergent between animals and timepoints without showing a possible pattern (Appendix A).

### 3.3. Diversity Analysis

#### 3.3.1. α Diversity

The α diversity allowed assessing the diversity of the microbiome present in a sample with a low value indicating a lower diversity and vice versa [20] (Appendix A). As in the previous section, the results were also dissimilar. Mostly, they were statistically significantly different, excluding those from the T1 and T2 samples obtained from Animal 3 (*p* = 0.512691) and Animals 2 and PC at T2 (*p* = 0.827259).

#### 3.3.2. β Diversity

The β diversity allowed comparing the bacterial populations present in the different faecal samples based on the Bray–Curtis dissimilarity index [20]. As observed for α diversity, no pattern was detected, but it is noteworthy to mention that the values for the samples from the NC animal taken in T1 resembled the diversity present in the capsules (Appendix A).

## 4. Discussion

This pilot study aimed to assess the influence of long-term Faecal Microbiota Transplantation in animals with chronic diarrhoea; therefore, the collection of the faecal samples to be analysed was performed at a later stage after transplantation, enabling the long-term evaluation of the intestinal microbiota adaptation process to this new stimulus.

Animals to be included lived in a kennel, which allowed their continuous vigilance by veterinary medical trained personnel throughout the assay. Furthermore, being a study about the gut microbiome, the selection of animals that shared the same environment, diet and veterinary care aimed at minimising the variables present [21] and standardising the sample collection and storage procedures. Together with the small sample size, distinctive characteristics presented by the animals, such as age, body condition and vaccination status, can be considered as limitations, as they can influence the constitution of the gut microbiota [21,22,23]. For example, the absence of information on the vaccination status of Animal 2 should be noted, since its samples had different results from the others. The absence of a detailed clinical history and of a diagnosis of the underlying cause of the change in faecal consistency of the animals under study should also be noted [11] as well as the characterisation of a possible chronic enteropathy [24]. Nevertheless, this study only aimed to be a preliminary evaluation of the influence of FMT based on the oral administration of lyophilised faecal capsules on the faecal consistency and on the composition of the intestinal bacterial population of the dogs under study.

If rigorous screening of faecal material is performed, side effects associated with FMT are rarely seen [6,25], which was also evidenced in this study. Regarding faecal consistency, the improvement of this parameter was evident in Animals 1 and 3 as well as the maintenance of the values in the long-term evaluation (T4). Also, Animal 2 showed a positive evaluation regarding this parameter during the FMT period, but in the long-term assessment, the faecal consistency of its samples relapsed, showing faecal consistency values similar to the one observed at T0. This animal was characterised as being very anxious and nervous [26] and, in addition, it was the youngest animal enrolled for the study and the one with an unknown vaccination status.

According to *AnimalBiome*^®^ [27], the tested capsules contain 20 × $10^{9}$ CFU per gram of faeces. The quantification performed in this study showed only a 100 CFU/g difference, revealing that the capsules’ manufacturing procedure supports the conservation of the bacterial population. Regarding capsules constitution, the results showed that the families *Lachnospiraceae*, *Clostridiaceae* and *Peptostreptococcaceae* of the class *Clostridia*, phylum *Firmicutes*, were the most abundant. However, the value corresponding to this phylum was higher than expected, resulting in extremely low concentrations of the remaining phyla [10]. In addition, the higher concentration of Actinobacteria is not in accordance with the study by Pereira and Clemente [28], who stated that this phylum is generally present in a lower concentration. The reduced concentration of Proteobacteria agreed with previous studies, which indicate that this phylum only increases in diseased animals [29]. Regarding the species present, the predominant one was *C. hiranonis*, the main bacterium responsible for the conversion of bile acids, with bacteria belonging to the genera *Blautia* and *Ruminococcus*, which also have a beneficial role in maintaining the host homeostasis, also presenting high concentrations. *S. luteciae* was also present at a high concentration, which is not in accordance with available publications, as the highest concentration of this bacterial group is usually associated with disease—particularly with chronic enteropathy [3,30]. *C. perfringens* concentration depicted the commensal presence of this bacterial species, not being indicative of gastrointestinal disease [10,31]. However, this species was present at a higher concentration than *F. prausnitzii*, which, according to Honneffer et al. [30], when present at a low concentration can be considered a marker of faecal dysbiosis.

In the faecal samples of the control animals, the phyla concentrations were in accordance with previous studies [10,28] except for the reduced concentration of Fusobacteria since, according to Niina et al. [29], a low proportion of this Phylum and an increase in Proteobacteria may be suggestive of an enteropathy. However, these animals did not show an increase in Proteobacteria or any clinical signs suggestive of a gastrointestinal disorder. Of note is the high concentration of Bacilli, more specifically of *Lactobacillus*, which is generally increased in the case of disease [2,32]. According to Aboim [33], the fact that the animals under study were housed in a kennel is a possible justification for this result. Additionally, *Clostridiaceae*, *Peptostreptococcaceae* and *Lachnospiraceae* were only at high concentrations in the NC animal at T1, which does not corroborate the presence of a microbiota in equilibrium [4]. However, most samples showed a low concentration of *Enterobacteriaceae* and *Streptococcus* spp. and a high concentration of *C. hiranonis* and *Blautia* spp., which are characteristic of a healthy microbiota [3]. As observed in the capsule’s composition, *F. prausnitzii* was also found in reduced quantities, and the commensal presence of *C. perfringens* was also detected. Regarding the T1 sample of the negative control, despite the animal’s normal physical exam and constant faecal consistency during the study, an increase in bacterial diversity was observed. However, it is important to say that healthy animals may present an unstable microbiota, which may justify this result [3,4].

Regarding the action of the capsules on microbiome composition, although no significant differences were detected, certain trends were observed that may be a result of FMT administration [4,34,35]. For example, in most faecal samples, there was an increase in Bacteroidetes and Actinobacteria and a decrease in Firmicutes, corresponding to a decrease in the predominant Phylum and an increase in the Phyla previously present in lower concentration, conferring an increase in the diversity of the bacterial population. Nevertheless, the opposite variations were observed in the samples from the PC animal at T2, which may be associated with the normalisation of the intestinal microbiota and adaptation to capsules ingestion. The slight increase in Actinobacteria in Animal 2 observed only after two months of FMT, which may reflect a higher degree of initial faecal dysbiosis, and the fact that the samples from the NC animal at T2 were the only ones that showed a lower concentration of Actinobacteria when compared to the one present in the samples collected at T0, reflect the importance of capsule administration in the increase in this phylum. The variation verified in Proteobacteria revealed an attempt to reduce these values; however, the values for this phylum were never particularly high in the animals under the study despite the association of Proteobacteria with an unbalanced microbiota [3]. Overall, the samples collected after FMT were characterised by the opposite variation. However, when compared to T0, the effect of the capsules was associated with a decreased concentration of Firmicutes and an increased concentration of Bacteroidetes. Actinobacteria concentration did not present significant variations. Finally, it is noteworthy that Animal 2 presented the highest value concentration of Proteobacteria, which may justify the worsening of the faecal consistency.

Regarding bacterial classes, in the first month, the concentration of Bacilli decreased while those from the other classes—Clostridia, Coriobacteriia, Erysipelotrichi and Bacteroidia—increased. In the second month, the variations differed between animals. After the end of FMT, Bacteroidia decreased in most animals, and the remaining classes showed divergent variation, which may be due to a possible dysbiosis which may trigger different responses in the post-FMT period. Despite this, the influence of capsules remained, since most samples showed a lower value of Bacilli and higher value of Clostridia, Erysipelotrichi and Bacteroidia when compared to T0, resembling that the ones present in a healthy microbiota [3,32]. Once again, Animals 2 and NC represented the exceptions, which can be justified by the differences presented by Animal 2 as well as by the fact that no capsules were administered to Animal NC. Only Coriobacteriia, like the phylum to which it belongs, did not show quite different values from those of from T0.

More specifically, during FMT, as expected [8,32], most faecal samples showed an increase in *Prevotellaceae*, *Lachnospiraceae* (*Blautia* spp.), *Peptostreptococcaceae*, *Clostridiaceae* (*C. hiranonis*) and *Faecalibacterium prausnitzii*, together with a decrease in *Lactobacillaceae* and *Enterobacteriaceae*, with the exception of Animals 2 and NC. Particularly, a decrease in *Streptococcaceae* was evident in all animals only in the second month of the FMT protocol except in Animal NC. This seems to indicate that an FMT period longer than 30 days may be necessary to cause a decrease in this family concentration.

In the month following the end of the FMT, most samples showed the opposite variation of the one observed after capsule administration. This minor relapse is consistent with the observation made by Chaitman and Gaschen [2] that dogs with chronic diarrhoea often necessitate multiple FMT treatments to prevent relapses. However, when compared to the results from samples obtained at the beginning of the study, most animals showed higher concentrations of *Prevotellaceae*, *Lachnospiraceae*, *Clostridiaceae* and *F. prausnitzii* and lower concentrations of *Streptococcaceae* and *Enterobacteriaceae*, which suggest an approximation with a healthy microbiota [10], again the exceptions being Animals 2 and NC. Moreover, minimal changes were observed regarding *Fusobacteriaceae* [10].

The β diversity revealed a high dissimilarity between the samples collected at T0 from all dogs and the capsules. At the end of the study, all animals showed a greater similarity with the samples taken at T0, which portrays, as mentioned earlier, a slight relapse. However, at T3, higher values of bacteria were considered beneficial, and lower values of disease-associated bacteria were also evident [36].

The lack of predefined reference intervals made it impossible to define the degree of dysbiosis present in each animal at the beginning of the study, and it was also not possible to determine the faecal dysbiosis index [37,38] due to the small sample size.

## 5. Conclusions

Nowadays, FMT is not widely used in the veterinary setting; however, this procedure could become a reliable alternative to the use of antibiotics. The implementation of FMT through oral capsules administered at home would highly facilitate the wider application of this technique.

In this study, it is noteworthy that no adverse effects were demonstrated, emphasising the safety of this alternative to antibiotics. Additionally, the administered capsule period and dosage were successful in restoring the faecal consistency of Animals 1 and 3. However, a similar restorative effect was not observed in the case of Animal 2. For this particular dog, an extended administration period or a treatment strategy addressing the root cause of the diarrhoea could be necessary. Lastly, considering the diverse characteristics of the small group of dogs included in this study, caution is advised when extrapolating the results to a larger population. Nevertheless, despite these limitations, noticeable trends persist within the findings, highlighting the potential of FMT interventions in canine patients.

In the future, it would be extremely relevant to conduct a prospective experimental project with a larger number of animals with similar characteristics and with a definitive gastrointestinal diagnosis. Furthermore, it would be pertinent to assess other parameters such as the faecal dysbiosis index to define the degree of initial dysbiosis and verify its evolution throughout the FMT.

In conclusion, it is important to investigate the benefits of FMT in veterinary medicine, aiming at developing guidelines for the standardised use of this therapy.

## Figures and Tables

**Figure 1 genes-14-01676-f001:**
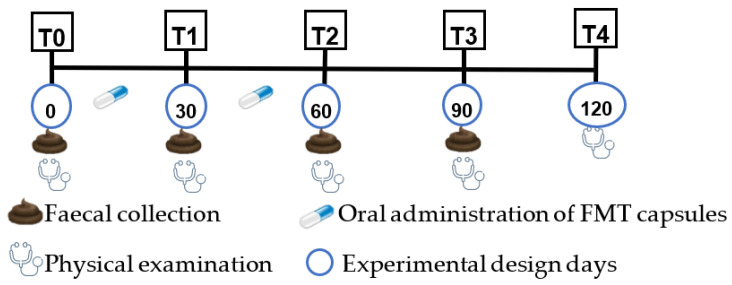
Representation of the different temporal moments of the study.

**Figure 2 genes-14-01676-f002:**
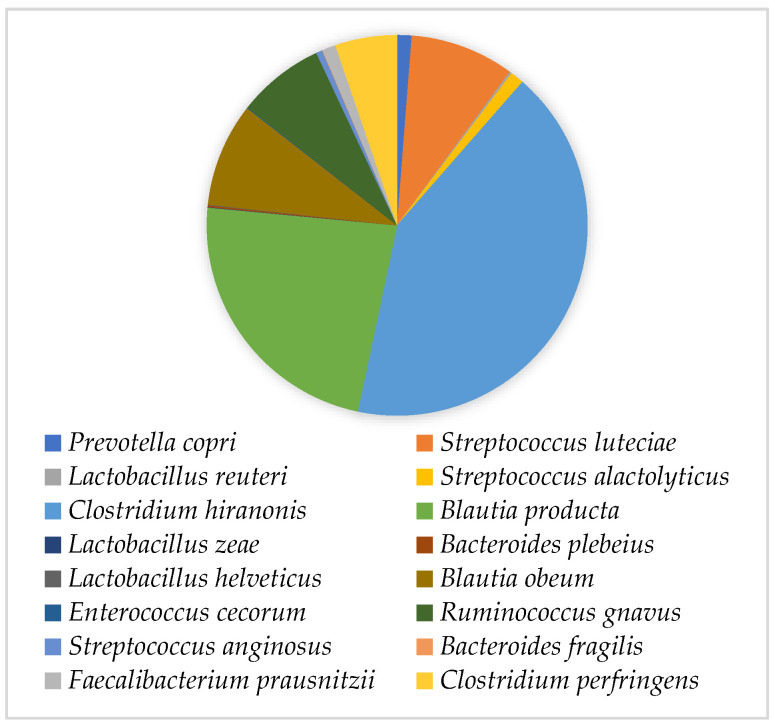
Bacterial species present in the faecal capsules used in this study.

**Figure 3 genes-14-01676-f003:**
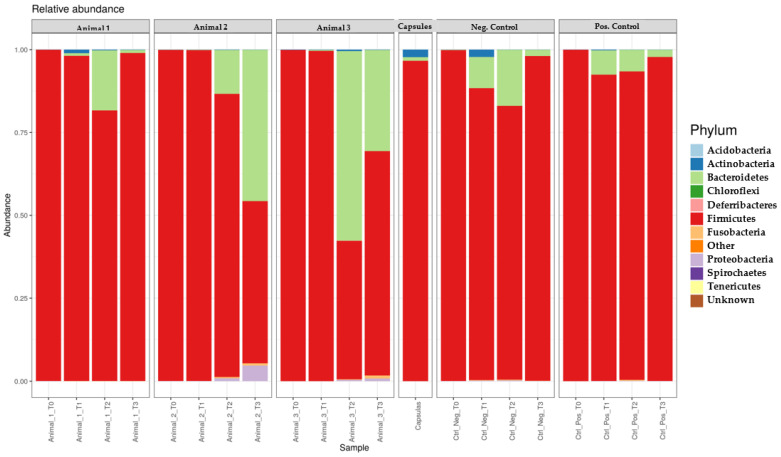
Relative concentration of each bacterial phylum present in each faecal sample during the study compared to the one present in the capsule and negative control faecal samples.

**Figure 4 genes-14-01676-f004:**
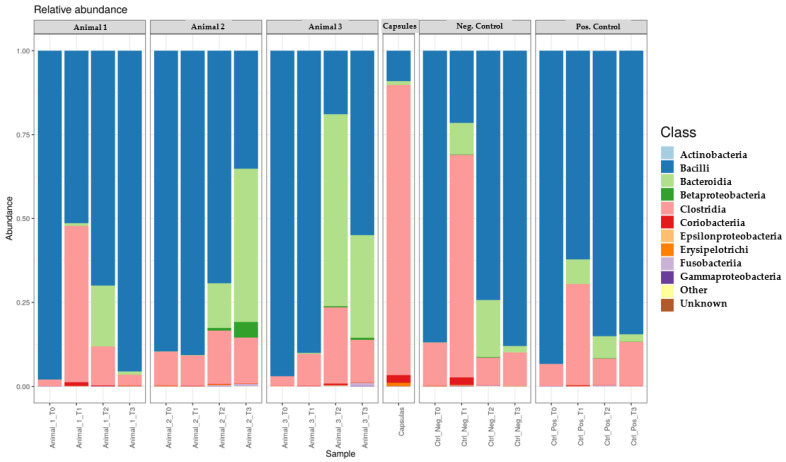
Relative concentration of each bacterial class present in each faecal sample during the study compared to the one present in the capsule and negative control faecal samples.

**Figure 5 genes-14-01676-f005:**
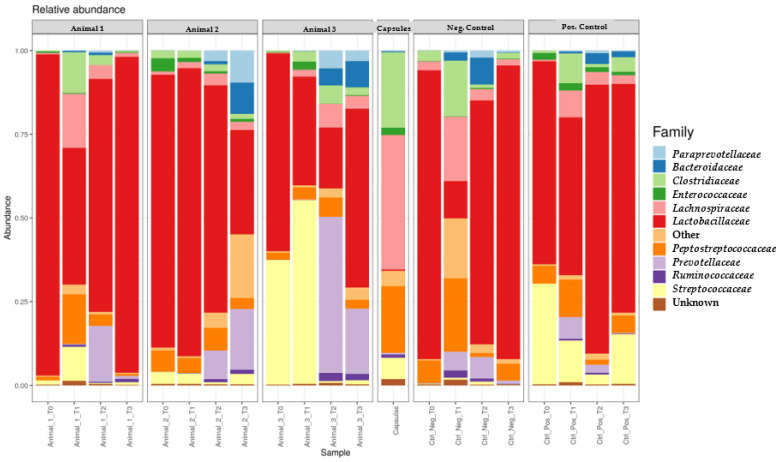
Relative concentration of each bacterial family present in each faecal sample during the study compared to the one present in the capsule and negative control faecal samples.

**Figure 6 genes-14-01676-f006:**
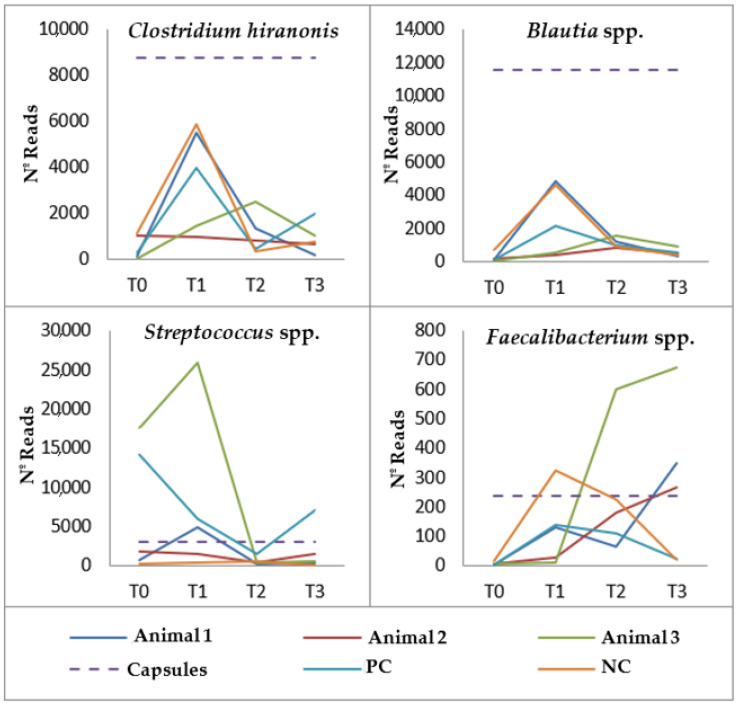
Variation in the concentration of *Clostridium hiranonis*, *Blautia* spp., *Streptococcus* spp. and *Faecalibacterium prausnitzii* in the different faecal samples analysed during each timepoint of the study.

**Table 1 genes-14-01676-t001:** Characterisation of the study animals.

	Sex	Neutered	Age (years)	Weight (kg)	Deworming	Vaccination	Faecal Consistency
**Animal 1**	Male	Yes	6	25	Regularised	Regularised	6/7
**Animal 2**	Male	Yes	2	25	Regularised	Not Regularised	7
**Animal 3**	Male	Yes	10/11	15	Regularised	Regularised	5/6
**Positive** **Control**	Male	Yes	2	10	Regularised	Regularised	4
**Negative** **Control**	Male	Yes	2	27	Regularised	Regularised	4

**Table 2 genes-14-01676-t002:** Faecal consistency presented by the different animals throughout the study, classified according to the Bristol Scale.

	T0	T1	T2	T3	T4
**Animal 1**	6/7	4	4	4	4
**Animal 2**	7	5	5	5/6	6
**Animal 3**	5/6	4	4	4	4
**Positive Control**	4	4	4	4	4
**Negative Control**	4	4	4	4	4

## Data Availability

The datasets used during the current study are available from the corresponding author on reasonable request.

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
