# Peer review of "Effect of the Administration of a Lyophilised Faecal Capsules on the Intestinal Microbiome of Dogs: A Pilot Study"

_genes, 2023, doi:10.3390/genes14091676_

Round 1

Reviewer 1 Report

The most important problem with this study is that low number of animals included completely precludes the chance to achieve scientifically sound results. Since there are only five dogs in the study (divided into three (!!!!) experimental groups) – it is hard to interpret the results obtained. With such a study design no statistic is possible. That is why even in the title of the paper it must be pointed out that this is a pilot or preliminary study. In fact, this is a description of three different cases of dogs treated with FMT. That is why I would avoid any generalization of the result obtained, especially that the dogs selected to the study were very different.  

But even writing about three different case studies requires more information about the background of the chronic diarrhea. Why at least 3 weeks of diarrhea were considered as definition of “chronic” process? Generally, diarrhea is a symptom and not a disease. If it is present longer than 3 weeks – the origin of the disease is probably still present within the body. But The Authors wrote that the animals were completely healthy, except the diarrhea. And the only marker of the disease was fecal consistency. In my opinion – this is simply not enough. What about digestive disorders, parasites etc.? How to interpret the result from Animal 2 when I simply do not know whether the primary disease was still present after 4 months of the experimental period or not. That is why the Authors must describe clinical status of every dog in detail, both before and after procedure. In the current paper FMT is considered as a treatment method so the indicators of health status change in dogs must be clear and unequivocal.

Completely different thing is the interpretation of changes in microbiota composition. Since there are not so many papers about FMT – it is hard to interpret it and compare with other research. That is why I think the paper is a valuable input into the common knowledge about the protocol of FMT and its results. I guess whether more information could be given about the composition and the source of FMT capsules. I know that it is a commercial product but on the other hand this is a scientific paper and at least the donor dog description (inclusion criteria to become a FMT donor) could be of value. 

To sum it up, in my opinion the paper is very descriptive and the only conclusion I have is the FMT was quite safe since there were no worsening of the clinical signs in any treated animal. 

Author Response

Dear Reviewer,

We would like to thank you for the comments and the discussion points presented. We agree that they will improve the quality and understanding of the manuscript. Please see below the response to each of your question. Also, all changes performed in the revised version of the manuscript were highlighted in yellow.

Point 1: The most important problem with this study is that low number of animals included completely precludes the chance to achieve scientifically sound results. Since there are only five dogs in the study (divided into three (!!!!) experimental groups) – it is hard to interpret the results obtained. With such a study design no statistic is possible. That is why even in the title of the paper it must be pointed out that this is a pilot or preliminary study. In fact, this is a description of three different cases of dogs treated with FMT. That is why I would avoid any generalization of the result obtained, especially that the dogs selected to the study were very different. 

Response 1: We recognize your question. In fact, we understand that the number of animals used in this study is small. However, this is an in vivo pilot study that, we believe, will help to develop larger studies in this thematic area, in the future. Other studies using microbiome analysis in dogs also used few animals in pilot studies, as you may see, for example, in the following article: DOI: 10.7717/peerj.11626. Considering this point and your suggestion we have changed the manuscript title to emphasize this work as a pilot study, to “Effect of the Administration of a Lyophilized Faecal Capsules on the Intestinal Microbiome of Dogs: A Pilot Study “, and we also added a new paragraph in the conclusion section about the limitation of the study population, as you may see in lines 606-608.

Point 2: But even writing about three different case studies requires more information about the background of the chronic diarrhea. Why at least 3 weeks of diarrhea were considered as definition of “chronic” process?

Response 2: We thank you for your question. The definition of chronic diarrhea used in this study is based on a well-known veterinary reference book (Reference 11). While it is acknowledged that certain clinics may interpret it differently, our decision to adhere to the definition provided in the attached reference is based on the aim to remain consistent with the available bibliography.

Point 3: Generally, diarrhea is a symptom and not a disease. If it is present longer than 3 weeks – the origin of the disease is probably still present within the body. But The Authors wrote that the animals were completely healthy, except the diarrhea. And the only marker of the disease was fecal consistency. In my opinion – this is simply not enough. What about digestive disorders, parasites etc.?

Response 3: We understand your question. Considering the animal’s health we refer in lines 202 and 203 that the animals presented no signs of disease, with the exception of alteration in faecal consistency in Animal 1, Animal 2 and Animal 3. Considering that the animals that participated in this study were housed in a official kennel, it was not possible to obtain a detailed medical history of the animals. Despite that, we have performed a full clinical exam of all animals prior to the study and during the timepoints of the study, as presented in figure 1. All animals in official kennels are submitted to regular vacination and deworming, following the national legislation ( see lines 103-104). Considering other digestive disorders, we focus on detect and classify the faecal consistence, in accordance to a well known scale (the Bristol Stool Scale, reference 12), and not in specific gastrointestinal disorders, considering the medical records of the participating animals.

Point 4: How to interpret the result from Animal 2 when I simply do not know whether the primary disease was still present after 4 months of the experimental period or not. That is why the Authors must describe clinical status of every dog in detail, both before and after procedure. In the current paper FMT is considered as a treatment method so the indicators of health status change in dogs must be clear and unequivocal.

Response 4: We understand your comment. Due to the origin of the animals and limited availability of resources, we were unable to explore the underlying primary ditsease or ascertain the cause of the altered fecal consistency. Our actions were confined to conducting a thorough physical examination, which is extensively detailed in the article. The Results section specifically says the results obtained “the physical examination of the animals’ revealed no mild or gross signs of disease”. Particularly, Animal 2 presents intriguing aspects due to its anxious behavior and recent arrival at the shelter. However, since the article is categorized as a pilot study, we believe it will emphasize the caution required when interpreting these results.

Point 5: Completely different thing is the interpretation of changes in microbiota composition. Since there are not so many papers about FMT – it is hard to interpret it and compare with other research. That is why I think the paper is a valuable input into the common knowledge about the protocol of FMT and its results. I guess whether more information could be given about the composition and the source of FMT capsules. 

Response 5: We thank you for your comment. As suggested, we have included more information regarding the FMT capsules, as you may see in lines 123 to 125 (“Regarding the composition of the capsules, in addition to the faecal material, these also contain inactive ingredients such as glycerol and the enclosing capsule, which confers resistance to enzymatic action within the gastric and intestinal compartments.”).

Point 6: I know that it is a commercial product but on the other hand this is a scientific paper and at least the donor dog description (inclusion criteria to become a FMT donor) could be of value.

Response 6: We agree with your comment and one new paragraph was added to the material and methods section as you may see in lines 114 to 122.

Point 7: To sum it up, in my opinion the paper is very descriptive and the only conclusion I have is the FMT was quite safe since there were no worsening of the clinical signs in any treated animal.

Response 7: Considering your comment, we added one sentence rephrasing about the safety of the protocol in the conclusion section (see lines 601-602).

Reviewer 2 Report

Dear Dr. Carapeto,

I hope this message finds you well.

I would want to express my gratitude for the work you put into the study that was presented in the manuscript that you submitted to Genes Journal. I understand the importance of your work in addressing the impact of Fecal Microbiota Transplantation and its possible impact on restoring the intestinal stability of dogs after GIT Dysbiosis.

I applaud you for your thorough study and straightforward presentation of the research findings. Your work reveals a thorough knowledge of the subject. I must, however, address issues with the research design since the number of dogs in each group was very small and may affect the results.

I encourage you to address the aforementioned points. Your work has the potential to significantly advance the field with the appropriate adjustments.

I want to thank you once more for your useful input and I'm looking forward to reading your feedback.

Sincerely,

Author Response

Point 1: Dear Dr. Carapeto,

I hope this message finds you well.

I would want to express my gratitude for the work you put into the study that was presented in the manuscript that you submitted to Genes Journal. I understand the importance of your work in addressing the impact of Fecal Microbiota Transplantation and its possible impact on restoring the intestinal stability of dogs after GIT Dysbiosis.

I applaud you for your thorough study and straightforward presentation of the research findings. Your work reveals a thorough knowledge of the subject. I must, however, address issues with the research design since the number of dogs in each group was very small and may affect the results.

I encourage you to address the aforementioned points. Your work has the potential to significantly advance the field with the appropriate adjustments.

I want to thank you once more for your useful input and I'm looking forward to reading your feedback.

Dear Reviewer,

We would like to thank you for the comments and the discussion points presented. We agree that they will improve the quality and understanding of the manuscript. Please see below the response to your question. Also, all changes performed in the revised version of the manuscript were highlighted in yellow.

Response:

We thank you for your comment and suggestion. In fact, we understand that the number of animals used is small. However, this is an in vivo pilot study that we believe it will help to develop larger studies in this thematic area, in the future. Other studies using microbiome analysis in dogs also used few animals in pilot studies, as you may see, for example, in the following article: DOI: 10.7717/peerj.11626. Considering this point and your suggestion we have changed the manuscript title to emphasize this work as a pilot study, to “Effect of the Administration of a Lyophilized Faecal Capsules on the Intestinal Microbiome of Dogs: A Pilot Study“ and we also added a new paragraph in the conclusion section about the limitation of the study population, as you may see in lines 606-608  (“Lastly, considering the diverse characteristics of the small group of dogs included in this study, caution is advised when extrapolating the results to a larger population.”).

Reviewer 3 Report

The manuscript titled ”Effect of the Administration of a Lyophilized Faecal Capsules on the Intestinal Microbiome of Dogs” aims to show the impact of a fecal microbial transplant on three dogs with chronic dysbiosis and two control dogs. The manuscript is fairly well written, but minor language revisions could be done. The methods used are sound although not all my area of expertise. The results have been reported extensively and with nice clarifying figures. There are some comments I think need to be addressed prior to publication.

Line 57-58: This sentence seems to be missing a word at the end.

Line 67: The “goal of the study was to develop a pilot study” sounds odd. Would it be easier to say that “the goal of this pilot study was…”?

Line 83: I have not seen “superior” used in this way. Could “longer” or “over” be used?

Line 85-86: Since the dogs shared the same environment, would it be possible for feco-oral transmission through pika? If so, would this not mix the groups and hence invalidate the study?

Line 113: Could the negative control animal have been given a placebo? If so, why was this not used? Also, here you could just talk about the negative control animal or dog, since the group had a n=1

Line 115-116: This is essentially the same information as on line 107-109, correct? Please simplify.

Materials and methods: For clarity, there should be information on whether the dogs were housed indoors or outdoors or both.

Line 138-143: Were these plates incubated aerobically, anaerobically, or both? If only one or the other this should be mentioned as a large proportion of the culturable bacteria may have been omitted. As shown by the NGS results, many bacteria were anaerobic.

Figure 3 and 4: Is there an explanation for the Ctrl_neg_T1 specimen, which is differs quite a bit from the other timepoints? There is something about the NC animal in the discussion, but I could not find text on this.

Line 486-489: This is more discussion than conclusion, correct?

Some minor language revision could be beneficial.

Author Response

Dear Reviewer,

We would like to thank you for the comments and the discussion points presented. We agree that they will improve the quality and understanding of the manuscript. Please see below the response to each of your question. Also, all changes performed in the revised version of the manuscript were highlighted in yellow.

Point 1: The manuscript titled” Effect of the Administration of a Lyophilized Faecal Capsules on the Intestinal Microbiome of Dogs” aims to show the impact of a faecal microbial transplant on three dogs with chronic dysbiosis and two control dogs. The manuscript is fairly well written, but minor language revisions could be done.

Response 1: We thank you for your comment. We have made some revisions as you may see all over the manuscript highlighted in yellow, and specifically in lines 17, 67, 82,104, 105, 106, 137, 139, 209, 230, 270, 272, 295, 337, 348, 349, 399, 479, 480, 481, 493, 494, 505, 571 and 590.

Point 2: Line 57-58: This sentence seems to be missing a word at the end.

Response 2: We agree with your comment and corrected the sentence (“However, it is known to contribute to the enrichment of the microbiome and to the alteration of microbial profiles”).

Point 3: Line 67: The “goal of the study was to develop a pilot study” sounds odd. Would it be easier to say that “the goal of this pilot study was…”?

Response 3: We agree with your point and corrected the sentense in accordance. ( “The goal of this pilot study was to evaluate the influence of the long-term administration of freeze-dried faecal capsules via oral route on the composition of the intestinal microbiota of dogs and their effectiveness in correcting animals’ faecal consistency.”).

Point 4: I have not seen “superior” used in this way. Could “longer” or “over” be used?

Response 4: We agree with your suggestion and changed the word (“the other three animals presented chronic diarrhoea (over or equal to 3 weeks) and constituted the target animals of the study (treatment group).”).

Point 5: Since the dogs shared the same environment, would it be possible for feco-oral transmission through pika? If so, would this not mix the groups and hence invalidate the study?

Response 5: We understand your question. Most of the animals that participated in this study were housed in different boxes, only Animal 1 and the negative control were sharing the same space. When we mentioned the same environment, we were referring to the water, food and air quality, as well as the cleaning procedures. In order to clarify this point, we changed this description, as you may see in lines 84 to 87 (“The selected dogs were housed in a kennel, in separate outdoor covered boxes (except for Animal 1 and the NC, which were together in the same box), shared a common environment (including water and air quality and cleaning procedures), and were fed with the same diet.)”

Point 6: Could the negative control animal have been given a placebo? If so, why was this not used?

Response 6: We thank you for this comment. The negative control (NC) animal did not receive any placebo, since the FMT capsules were a commertial product without any specific delivery system or compound excipient within, so it was not possible to construct a reliable placebo.

Point 7: To sum it up, in my opinion the paper is very descriptive and the only conclusion I have is the FMT was quite safe since there were no worsening of the clinical signs in any treated animal.

Response 7: We agree with your comment, and we included one new sentence in the conclusion section about the safety of the protocol (see lines 601-602).

Point 8: Also, here you could just talk about the negative control animal or dog, since the group had a n=1.

Response 8: We agree with your point and corrected the sentence in lines 137-138 (“The negative control animal was not submitted to any treatment or placebo.”).

Point 9: Line 115-116: This is essentially the same information as on line 107-109, correct? Please simplify.

Response 9: We understand your comment and proceeded with a simplification of the sentence, as you may see in lines 128 to 131 (“This pilot clinical trial was a controlled study that included one dog as a negative control (with normal faecal consistency and no treatment) and two test groups: one with a single animal showing normal faecal consistency (positive control), and another group with three animals demonstrating altered faecal consistency (treatment group).”).

Point 10: Materials and methods: For clarity, there should be information on whether the dogs were housed indoors or outdoors or both.

Response 10: We agree with your comment and included some new information regarding the housing of the animals, as you may see in lines 84-87 (“The selected dogs were housed in a kennel, in separate outdoor covered boxes (except for Animal 1 and the NC, which were together in the same box), shared a common environment (including water and air quality and cleaning procedures), and were fed with the same diet.”).

Point 11: Line 138-143: Were these plates incubated aerobically, anaerobically, or both? If only one or the other this should be mentioned as a large proportion of the culturable bacteria may have been omitted. As shown by the NGS results, many bacteria were anaerobic.

Response 11: Thank you for your comment. The quantification of the bacterial population present in the capsules was performed by incubating the plates aerobically. This procedure was performed to evaluate the viability of the FMT capsules, specifically to verify if culturable bacteria were present in the capsules. The microbial evaluation of the capsules was completed using genomic evaluation as described in lines 175 to 189, which allowed us to identify all bacteria present in the capsules. Considering this point, we included this information in line 172.

Point 12: Figure 3 and 4: Is there an explanation for the Ctrl_neg_T1 specimen, which is differs quite a bit from the other timepoints? There is something about the NC animal in the discussion, but I could not find text on this.

Response 12: We thank you for your comment. In fact, the negative control animal showed a change in composition in the T1 sample, characterized by a significant increase in the Clostridia class, followed by a subsequent decrease in the Bacilli class, and an increase in the diversity of bacterial families. Although this animal has showed a normal physical exam and a constant faecal consistency during the study, it is important to say that healthy animals may present an unstable microbiota, which may justify this result. Considering this point, we included new information in the discussion section, as you may see in lines 534 to 538.

Point 13: Line 486-489: This is more discussion than conclusion, correct?

Response 13: We agree with your comment and corrected the sentence as you may see in lines 580-581 (“This minor relapse is consistent with the observation made by Chaitman and Gaschen that dogs with chronic diarrhoea often necessitate multiple FMT treatments to prevent relapses.”).

Round 2

Reviewer 1 Report

I accept the changes in the manuscript. In this form this should be considered as pilot study showing no adverse effect of investigated commercial product with FMT in three dogs with chronic diarrhea.

Regards